# Socio-Economic Factors for Anthill Soil Utilization by Smallholder Farmers in Zambia

**Kafula Chisanga** [1,2,*]**, Ernest Mbega** [1] **and Patrick Alois Ndakidemi** [1,2]

[1]  Department of Sustainable Agriculture, The Nelson Mandela African Institution of Science and Technology, P.O. Box 447, Arusha 23311, Tanzania

[2]  Centre for Research, Agriculture Advancement, Teaching Excellence and Sustainability (CREATES) in Food and Nutrition Security, Arusha 23311, Tanzania

*  Correspondence: kafulac@yahoo.co.uk; Tel.: +255-783-640-428

**Abstract:** In this study, we surveyed two districts of Zambia—Choma and Pemba. The aim of this study was to obtain the perspective of farmers on anthill soil utilization practices for key information that could contribute towards the development of an anthill soil based research agenda. The study employed both a qualitative and quantitative method approach to gather data from the respondents, which included farmers and key informants. Qualitative data was analyzed using the triangulation method and Computer Assisted Qualitative Data Analysis Software (CAQDAS), Nvivo version 10, while data generated from quantitative interviews with a smart phone Application (Open Data Kit) were analyzed using Statistical Package for Social Sciences (SPSS). Results revealed that the key hurdles to the utilization of anthill soil lay in agro-climatic, biophysical, technological, land and institutional constraints. Broadly, farmers reported poor rainfall patterns (95%), decreasing soil fertility (70%), limited farm products (69%), finance (66%), limited access to research and extension services (55%) and security of land tenure (48%) as major constraints. We therefore advocate for strengthenedinstitutional linkages between research and extension for information dissemination, which would aid in decision-making used to promote integrated soil fertility management for improved agriculture production and productivity of rural households.

**Keywords:** anthill soil; practice; smallholder farmers; sub-Saharan Africa; technology; Zambia

## 1. Introduction

Food and Agriculture Organization and and Intergovernmental Technical Panel on Soils (2015) assert that soils are fundamental to life on Earth, but human pressures on soil resources are reportedly reaching critical limits. Therefore, prudent soil management is a vital element of sustainable agriculture and additionally provides a beneficial lever for climate regulation and a route for safeguarding ecosystem services and biodiversity as a whole. Considering that around 33% of the global soils and 40% of the soils in Africa are already degraded, a special focus on the restoration of degraded soils and maintenance of soil health is indispensable [1]. In sub-Saharan Africa, it is known that low soil fertility is one of the greatest biophysical constraints hindering agricultural production [2,3]. Soil fertility decline is linked to a diverse simultaneous degradation processes building on each other to produce a downward spiral in productivity and environmental quality. For example, [4] reported that the integrated effects of tillage and sub optimal applications of nutrient and organic matter undoubtedly leads to a reduction in soil organic matter. This consequently reduces the retention of essential plant nutrients, breaking down the physical structure of soil and in turn diminishing water infiltration and the water storage capacity of the soil. Additionally, most smallholder farmers face other hurdles of degradation processes such as erosion, salinization and acidification. Soil fertility decline encompasses

a myriad of variables including nutrient deficiency, physical and biological degradation, inappropriate crop varieties, cropping systems, pests and diseases. It also relates to an association between poverty and land degradation, unreasonable national and global incentive policies, and institutional failures. The degradation of soil fertility results in variant human and environmental problems among which low crop production, productivity and malnutrition serve as good examples.

In an effort to enhance their soil fertility challenges, many communities in Malawi, Niger, Sierra Leone, South Africa, Tanzania, Uganda, Zambia and Zimbabwe use anthill soil as a viable low risk endogenous soil fertility management strategy for crop production and food security [5,6]. In recent times, it has become evident that anthill soil utilization is a common practice in maize crop production amongst the smallholder farmers in Southern Zambia [7]. The practice reportedly entails digging, heaping and spreading the soil on the field. Anecdotal evidence suggests that maize and other crops planted and fertilized with anthill soil has proved to be with high vigour and gives a relatively high yield [8]. Many factors may have prompted farmers to use anthill soil in their agriculture production and the major reason could be high costs associated with an inorganic fertilizer—which is beyond their reach.

However, from a soil science point of view, various scientists have revealed that termites– including other fauna species in the soil environment—play a crucial role in anthill construction. The activity entails anthill building ants assembling woody debris for their nests and forage, which is utilized as food for colonies. As a consequence of this action, anthills are supplied with soil organic matter and inorganic nutrient elements, consisting of P, K, Ca, Mg and Na, in comparison with adjacent soils [9]. Further, bioturbation activities potentially contribute to changing; (i) soil physical characteristics, such as porosity and infiltration, (ii) soil micro-organisms and resulting fauna biomass and (iii) speedy of organic matter decomposition. Scholars have all linked soil physical-chemical characteristics of anthill soil to the building activities of ants [10,11] which makes anthill soils more fertile with key nutrient elements needed for crop production in comparison with surrounding soils.

Additionally, farmers have also indicated that once the anthill soils are applied in their agriculture lands, fertilizer application may not be required for at least three years [12]. Such suggestions by farmers need to be examined from a broader perspective. We therefore conducted the present study with a focus on answering the following threefold objectives; (i) to understand the status of anthill soil utilization at smallholder farm level in Choma and Pemba districts of Southern Zambia; (ii) to determine key benefits and constraints to the application of anthill soil practices in cropping systems and; (iii) to identify key information gaps that will contribute towards the development of the anthill soil based research agenda under conservation agriculture in Zambia and elsewhere in Africa. We envisage that the generated information in our work will provide insights on the existing situation regarding anthill soil utilization in the study areas for advising policy. To date, there is limited information on anthill soil utilization for Southern Zambia that can be used and integrated in conservation agriculture programmes for enhancing production and productivity at smallholder farmer scale—more so for the staple crop, maize, whose yield currently stands at less than 2 t ha$^{-1}$ [13].

The conceptual framework illustrating the link between the causes of low crop yield in an agricultural system at smallholder farmer level and how low alternative inputs contribute to improved soil fertility and effects of the adopted practices on soil environment and productivity is illustrated in Figure 1. The soil practices in our study focus principally on the anthill soil utilization in maize cropping systems as an endogenous soil fertility management practice by smallholder farmers in Zambia.

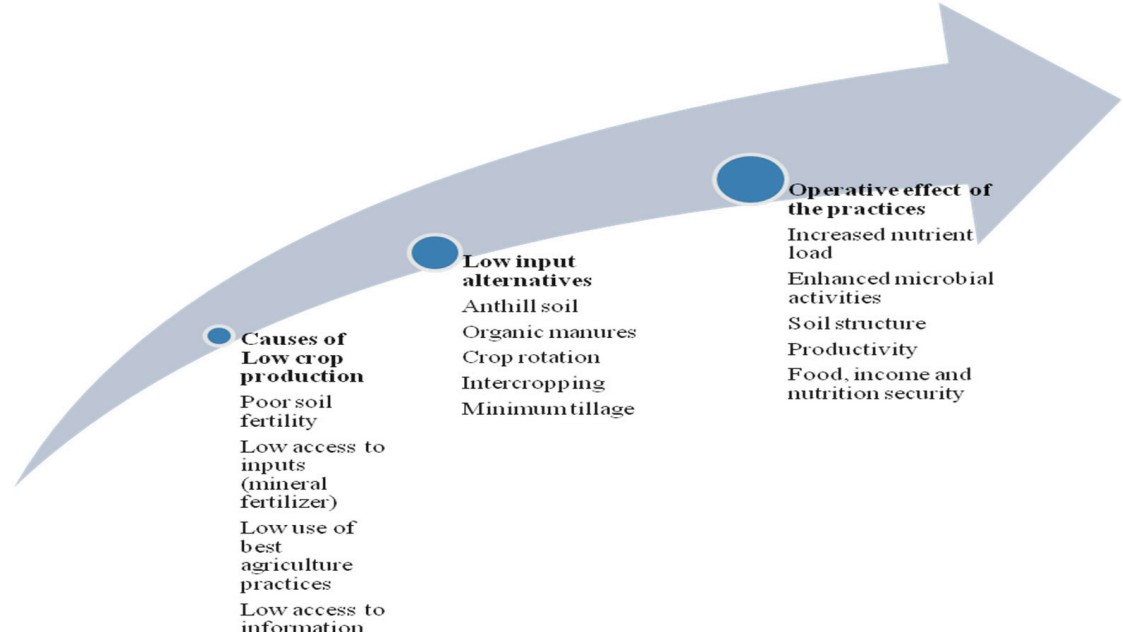

**Figure 1.** Relationship between certain low input soil management practices and effects on productivity and farmer livelihood. Adapted from Koopmans and Smeding [14] with modifications.

## 2. Materials and Methods

### 2.1. Research Sites

We chose two districts in Zambia for the study that included Choma and Pemba (Figure 2). The host districts were picked on the premise of their agro-ecological and farming systems context, which are more representative of Southern Province of Zambia. Choma district forms the hub of the Province because of its central location, but to a greater extent due to its important role in the economy of the province and the Republic of Zambia. The district lies approximately within longitudes 26°30′ and 27°30′ east of Greenwich, and latitudes 16° and 17°45′ south of the equator on the plateau of Southern Zambia.

The district is rural, and agriculture is the main form of income for the households. The Pemba district, on the other hand, is situated on the eastern side of Choma. The geographical coordinates for the district are 16°32′0″ South and 27°22′0″ East, respectively. Similar to Choma, the major activity of the people in the area is agriculture, which forms the spine to the economic activities of the district. Anthill soils in the study districts are proven as an alternative to commercially available fertilizers in cases where the commodity becomes difficult to access. Literature indicates that anthill soils have elevated levels of pH, organic matter, calcium, magnesium and phosphorus in comparison with surrounding soils, which are key aspects for enhanced crop development [15].

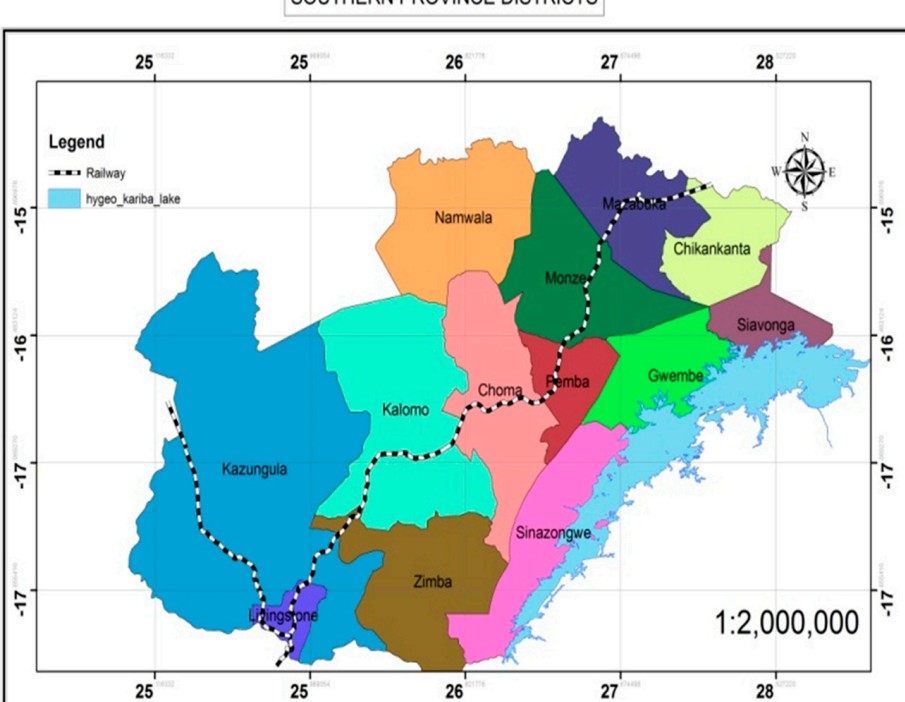

**Figure 2.** A map showing location of Choma and Pemba districts in Southern Province, Zambia Source [16].

### 2.1.1. Climate

The Choma district is located at 1400 m above sea level. The climate of the district is distinguished by temperatures that range from 14 to 28°C while the hours of sunshine range from nine to 12 hours in a particular day. The highest temperatures are normally experienced from October to December. However, during the rainy season, temperatures fall considerably.The lowest temperatures are normally experienced in the months of June and July. The rainy season potentially begins in October until the start of April. Additionally, the highest rainfall is recorded in January. Mean yearly rainfall for the district range from 800 to 830 mm—of which 369 mm occurs from January to February. Rainfall variations are however experienced every season and consequently lead to variable crop yields registered in the district [17].

### 2.1.2. Agriculture

This sector is the pillar of the rural economy in Choma and Pemba districts. CSO [18] approximates the overall number of agricultural households for Choma district to be around 165,589, out of which 142,659, are male and 22,930 are female, while in the Pemba district the gross agricultural households are 22,268, of which 11,735 are male, while 10,533 is accounted for by females [19].

The districts are predominantly maize growing, coupled with the high practice of anthill soil utilization due to poor soils or lack of capacity by some farmers to purchase adequate industrial fertilizer [7]. Choma is located in agro-ecological zone (AEZ) II and receives low to medium rainfall (600–800 mm), and farmers in the region are involved in ox-drawn-maize systems. On the other hand, Pemba—similar to Choma—is located in AEZ II, where occasional high rainfall is 800 mm (maximum) with farmers mostly using a hand hoe/ox-drawn-maize based farming system.

### 2.1.3. Demography

According to CSO [18], the Choma district population is estimated at 247,860, of which the male population is 121,451 while that of the female is 126,409. The annual growth rate for the district is estimated to be 2%. However, the Pemba district has an estimated population of 67,187, where 32,250

are male and 34,937 represent the females. The population was projected to increase to 77,938 by 2017 at an annual estimated growth rate of 1%.

## 2.2. Research Process

In order to respond to the stated research objectives, we conducted the present study in three stages following a chronological manner. Firstly, the research team held key informant interviews ($n = 6$) with public and private key decision makers from government departments and non-governmental organizations, to obtain their views regarding their knowledge on the status of anthill soil utilization and reasons for its use. Secondly, focus group discussions ($n = 4$) were held with farmers with the aim of obtaining an insight into their experiences using anthill soil technology, and lastly a quantitative data collection exercise was undertaken from the community ($n = 390$) to obtain data from variables of interest which formed the basis for data analysis. This mixed method approach was adopted as it offered an opportunity to obtain strong collective views from both quantitative and qualitative research [20].

The survey questionnaire was therefore organized to cover questions on household demographic characteristics, social, physical, human and natural assets, income sources, site characteristics, integrated soil fertility management practices and anthill soil utilization. The focus was on the year when the households started using the resource, social economic and biophysical factors which drove the households to start using the anthill soil, crops on which the resource was applied, benefits and challenges observed compared with conventional soil fertility management practices. Further, indicators were used to identify the suitability of anthills for soil improvement and management calendar in terms of the month when anthill identification, digging, transportation and application was carried out and by whom (either men, women, both or hired labour). Additional data were collected on how households accessed agriculture technology information for enhancing their crop production.

### Ethical Considerations

Before commencement of the present study, we obtained an informed consent through the camp agriculture extension officer of the research sites at the time of recruitment of the participants to the study, by reading and interpreting the designed consent form. This is in line with [21], who reported that oral consent has an advantage over written consent as it includes all respondents of any literacy scale. The informed consent spelt out the purpose of the study and any likely benefits and disadvantages which were to be accrued. This research was permitted by the IRB (Approval No. 2018-Feb-046).

## 2.3. Sampling Strategy

### Household Sampling

In this study, we employed a systematic farmer sampling strategy by village based on Agricultural Camp Register with the aid of the Cochran method illustrated in [22], to determine the sample population using the formula; $N = [Z^2 \, (p) \, (1 - p)/C^2]$; where $Z = 1.96$, $p = 0.5$ (% picking a choice response), $C$ = confidence interval (0.95). This provided us with approximately 390 respondents across the two study districts, whereby four agricultural camps per district with 49 farmers were interviewed on a structured questionnaire, using Open Data Kit (ODK) application installed on a smart phone. This resulted in a total of 196 surveys per study district. From the sampled population, 30% was reserved for women to engender the research process [23]. However, in this study, we only managed to capture 14% of active women involved in anthill soil technology practice.

Prior to data collection, we recruited 10 enumerators and trained them to help with administering the questionnaire using the ODK Application. Pre-testing of the digitized questionnaire was done to check for clarity, performance and ensure reliability. The timing of data collection was selected to coincide with the end of the harvest period when most farmers spend less time on agricultural activities in the field.

## 2.4. Data Analysis Strategy

We analyzed qualitative data using the triangulation method and Computer Assisted Qualitative Data Analysis Software (CAQDAS), Nvivo version 10 while data from structured questionnaires were subjected to the Statistical Package for Social Sciences (SPSS). Origin Pro 9.0 software (www.originlab.com) and Microsoft Excel (version 2010) were used for graphical representations and special computations. Chi-Square ($\chi^2$) test [24,25] was used to determine at 95% confidence interval any association between smallholder farmers' start year of using anthill soil to improve soil fertility and access to credit facilities and use of mineral fertilizer as well. The multinomial logit model [26] was further applied to analyze determinants of smallholder farmer's selection of anthill soil utilization in their agriculture crop production systems. The model adopted for anthill soil utilization determinants with possible conditional probabilities was specified as:

$$P(Y = 0|x) = \frac{1}{1 + e^{g_1(x) + \dots + g_{c-1}(x)}},$$

$$P(Y = 1|x) = \frac{e^{g_1(x)}}{1 + e^{g_1(x) + \dots + g_{c-1}(x)}},$$

$$P(Y = c - 1|x) = \frac{e^{g_{c-1}(x)}}{1 + e^{g_1(x) + \dots + g_{c-1}(x)}},$$

where $Y$ is an outcome and possible $c$ value of $(0, 1, \dots, c - 1)$; $Y = 0$ reference category and $x = (x_1, x_2, \dots, x_n)$ = set of independent variables. Logit of category $j$ against the baseline category was in the following form where $j = 1, 2, \dots, c - 1$:

$$g_{j\,(x)} = \ln\left[P(Y = j|x]/P(Y = 0|x) = \beta_{j0} + \beta_{j1}x_1 + \dots + \beta_{jp}x_p,\right.$$

$\beta_{j0}$ is the intercept; $\beta_{j1}x_1$, $\beta_{jp}x_p$ are coefficients denoting independent variables determining use of anthill soil.

## 3. Results and Discussion

### 3.1. Household Demographic Characteristics

Females accounted for 14% of respondents and males accounted for 86% of respondents. Of the entire households interviewed, 70% were married monogamy, 20% married polygamy, 5% widowed, 3% bachelors, 2% divorced, 1% spinsters and 0.30% separated. The demographic composition of anthill soil farmers from the study districts is indicated in Table 1. Naturally, the size of a household has a huge influence on a farmer's choice of livelihood activities and contributes to income disparities. For instance, an investigation on farming communities in the Democratic Republic of Congo [27], found that the marital status, family size and household age, influenced the income at a household level resulting from the communities involvement in diversified livelihood strategies. From the foregoing we note that most of the respondents in the current study were married and this contributed to making the choice for engaging in anthill soil utilization for crop production. Fang et al. [28] argued as well that the greater the human capital at household level, the more likely the farmers would choose agriculture as their main livelihood.

### 3.2. Social Assets

The majority (84%) were affiliated to social groups. Results indicated that 71% of the respondents were associated with farmer cooperatives and this was key to accessing new information on farming practices including other livelihood strategies. Others reported many conflicts (2%), having no money (2%), no benefits accrued (1%) and other reasons (6%) as the contributing factors for not being affiliated to any social organization. Membership to social groups is important, especially for

resource-constrained (poor, illiterate) smallholder farmers in technology adoption as this is a pathway for accessing high standard inputs (fertilizer and improved germplasm) including credit schemes [29].

**Table 1.** Overall household demographic characteristics across the study districts.

| Variable | Mean (*n* = 390) | Median (*n* = 390) | Mode (*n* = 390) | Std |
|---|---|---|---|---|
| HH size (number of individuals) | 9.00 | 8.00 | 8.00 | 3.20 |
| HH Age (age of individuals) | 44.00 | 43.00 | – | 12.60 |
| Number of male <14 years old | 2.40 | 2.00 | 1.00 | 2.03 |
| Number of female <14 years old | 2.00 | 2.00 | 1.00 | 1.80 |
| Number of male 15–49 years old | 2.00 | 2.00 | 1.00 | 1.60 |
| Number of female 15–49 years old | 2.00 | 2.00 | 1.00 | 1.40 |
| Number of male >50 years old | 0.30 | 0.00 | 0.00 | 0.50 |
| Number of female >50 years old | 0.30 | 0.00 | 0.00 | 0.70 |

NB: *n* = number; HH = household; std = standard deviation.

### 3.3. Physical Assets

Mean numbers of agriculture physical assets owned by respondents in the form of livestock included; cattle (7), goats (9), poultry (19), pigs (1), sheep (1), and other livestock (4), while observed mean values for agriculture equipment accounted for cultivators (1), axes (3), ploughs (1) and other agriculture assets (2). For non-agriculture assets, the average number for beds was (3), radio (1), bicycles (1) while for other non agriculture assets the mean was 1.

In the current study, we observe that livestock in form of poultry, goats and cattle are the major physical assets owned by the respondents coupled with some agriculture equipment such as ploughs, cultivators and oxcarts. The availability of these assets most likely contributed to the diversification of livelihood activities of the communities in the study districts. Hua et al. [30] stressed that ownership of physical assets is an incentive that drives communities to choose non-agricultural livelihoods and thus achieve transformed livelihoods.

### 3.4. Financial Assets

This study probed income patterns for two agriculture seasons; 2015–2016 and 2016–2017, respectively. Most anthill soil farmers indicated that their source of income in the period was accrued from sale of rain seed food crops (USD 382), sale of rain seed cash crops (USD 163) and trading (USD 143) (Table 2). Pender et al. [31] stressed that the different income strategies smallholder farmers are engaged in, have a possible direct effect for the outcomes they are interested in, which also affects them indirectly when a decision is made regarding adoption of the technology and land management practices.

**Table 2.** Overall income (USD) sources across the study districts.

| Income Source | 2015/16 Season | | | | 2016/17 Season | | | |
|---|---|---|---|---|---|---|---|---|
| | Mean (*n* = 390) | Median (*n* = 390) | Mode (*n* = 390) | Std (*n* = 3 90) | Mean (*n* = 390) | Median (*n* = 390) | Mode (*n* = 390) | Std |
| Trading | 143 | 0.00 | 0.00 | 382.3 | 125 | 0.00 | 0.00 | 275 |
| Gardening activities | 111 | 40 | 0.00 | 215 | 104 | 40 | 0.00 | 204 |
| Local chicken rearing | 69 | 20 | 0.00 | 175 | 64 | 15 | 0.00 | 258 |
| Goat rearing | 58 | 0.00 | 0.00 | 140 | 47 | 0.00 | 0.00 | 170 |
| Cattle rearing | 107 | 0.00 | 0.00 | 440 | 90 | 0.00 | 0.00 | 376.3 |
| Remittances | 13 | 0.00 | 0.00 | 79.3 | 12 | 0.00 | 0.00 | 53.1 |
| Sale of rain seed food crops | 382 | 0.00 | 0.00 | 1705 | 270 | 50 | 0.00 | 685 |
| Rain seed cash crops | 163 | 15 | 0.00 | 768 | 195 | 0.00 | 0.00 | 1060 |
| Piece work | 32 | 0.00 | 0.00 | 92 | 29 | 0.00 | 0.00 | 90 |
| Charcoal sale | 28 | 0.00 | 0.00 | 309 | 25 | 0.00 | 0.00 | 207 |
| Other sources | 44 | 0.00 | 0.00 | 400 | 47 | 0.00 | 0.00 | 320 |

NB: *n* = number; HH = household; std = standard deviation.

### 3.5. Natural Resource Assets

Land Resources

Based on the findings, our study observes that the respondents owned land for cultivation of various crops. However, 48% reported having a challenge regarding the security of land tenure while a further 41% acknowledged limited access to land as one of the land constraints faced on their farms. Most of the land in the study areas falls under the customary law, where the traditional leader has the authority and power of land distribution in the community. Regarding this, Pender et al. [31] reflected that property rights and the type of land tenure possessed by the smallholder farmer, affect land management and productivity for various reasons. For instance, if there is insecurity of tenure, the household operating the plot may have less incentive to invest in land improvement.This may not be the case, however, if the smallholder invests in the land itself, which may lead to increased tenure security [32]. In the study areas scenario however, there may be more investment in the land with insecure tenure.

### 3.6. Farmers' Perception of Sites' Characteristics

As noted by Waldman et al. [33], the texture, structure, and physical characteristics of the soil in our study areas (Choma and Pemba) are varied and on general scale, have poor physical properties with low nutrients dominated by sandy soils. This was no different from the findings in this study, wherebyin all sites across the districts, the major soil type as described by respondents was sandy (49%), clay (16%), loamy (15%) and a mixture of sandy loamy (12%), sandy clay (4%) and clay loamy (4%) respectively. Figure 3 shows the distribution of the soil type across the study districts as observed by smallholder farmers. Given the major soil type in the study areas, it may be one of the reasons attributed to most smallholders opting to use anthill soils as an alternative to commercially available fertilizers for soil fertility enhancement required to boost their crop yields.

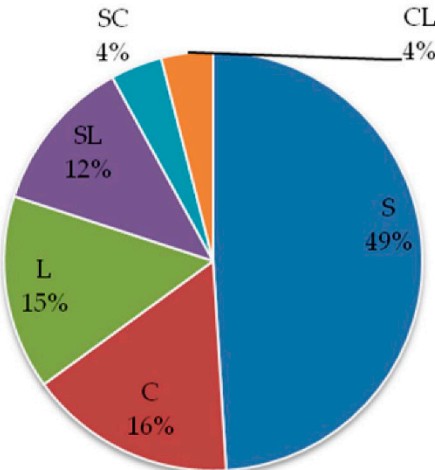

**Figure 3.** Site characterization of the study areas. (NB: S = sand; C = clay; L = loamy; SL = sand loamy, SC = sand clay, CL = clay loamy).

### 3.7. Application of Integrated Soil Fertility Management, Concepts and Principles by Farmers

Fairhurst [34] defined integrated soil fertility management (ISFM) as a group of soil fertility management practices which involve the use of industrial fertilizer, organic inputs and appropriate germplasm coupled with knowledge on how to adapt these practices to local environment, aiming at optimizing agronomic use efficiency, of the applied nutrients and enhancing crop productivity.

In this study, we noted that most of the smallholder farmers practiced ISFM concepts and principles in their agriculture production, ranging from crop rotation, manuring and intercropping to organic and inorganic inputs application (anthill soil and mineral fertilizers). Overall statistics (Table 3) indicated

that crop rotation of cereals with groundnuts was highly practiced (59%), followed by cowpeas (53%), beans (21%), Bambara nuts (5%), soybeans (4%), pigeon peas (1%), velvet beans (1%) and others (14%). Despite smallholder farmers practicing crop rotation in the study districts, it was noted however that the area rotated between legume and cereal crops was not equal. This may have been attributed to lack of enough legume seed by farmers and limited knowledge regarding crop rotation practice.

**Table 3.** Commonly rotated crops with cereals across the study districts.

| Crop | Count | Percentage (%) |
|---|---|---|
| Groundnuts | 228 | 59 |
| Cowpeas | 208 | 53 |
| Beans | 81 | 21 |
| Other specify | 54 | 14 |
| Bambara nuts | 18 | 5 |
| Soybeans | 17 | 4 |
| Pigeon peas | 3 | 1 |
| Velvet beans | 2 | 1 |

For other ISFM practices, we observed in this study that a number of sampled households across the districts used inorganic fertilizers (96%), cattle manure (84%) and practiced intercropping (50%) as part of enhancing soil fertility.

As for the fertility management practices employed, a majority of the respondents (Figure 4) indicated that they generated the organic matter by using various anthill soil application methods involving heaping and spreading on flat field (89%), placement in ripped lines (14%), practicing crop rotation (12%), placement in potholes/basins (10%), mixing with cattle manure (8%), covering soil with crop residues (4%), intercropping (3%), spot application (2%) and other methods (1%). This finding corroborates with the study by Acheampong et al. [35] in Ghana who pointed out that farmers normally engage in some form of supplementary land management actions that help to rejuvenate the soil fertility. In agreement to this assertion, Ansong et al. [36] reflected in Ghana as well that farmers opt to use organic based materials such as cattle manure, crop residues, land rotation among others, as a way of indigenous ISFM land management system.

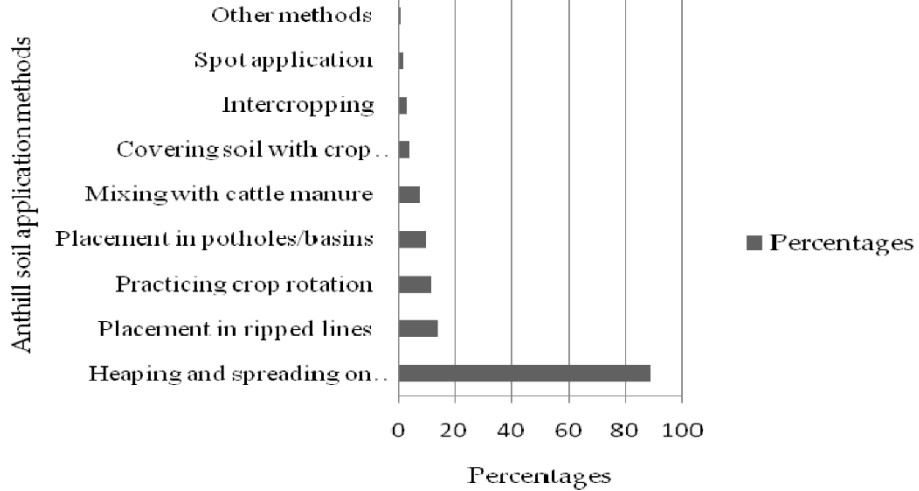

**Figure 4.** Comparison of anthill soil application methods.

In our study, it appears that the application of anthill soil on the flat field is prevalent. This may however prove to be wasteful as the ideal approach would be placement of anthill soil in ripped lines or basins. This method may be more efficient and less laborious if embraced by smallholder farmers.

*3.8. Status of Anthill Soil Utilization in Crop Production*

Overall, we observe in this study that most of the smallholder farmers started using anthill soil in crop production three years ago (32%) and those with five years and more than 10 years ago usage of the resource stood at 36% and 14%, respectively (Figure 5). A number of factors prompted the smallholder farmers to engage in the practice. Amongst the social economic and biophysical variables was the variable of limited finance to purchase mineral fertilizer (90%) and soil factors (95%). All the maize varieties (99%) were being grown under anthill soil treatment, while a paltry (1%) was not. The major improved maize varieties reported to be commonly grown under anthill soil amendment included; PANNAR 53 (55%), SeedCo 513 (27%) and ZamSeed 606 (22%), respectively. All these varieties are early to medium maturity (130 days to maturity maximum).

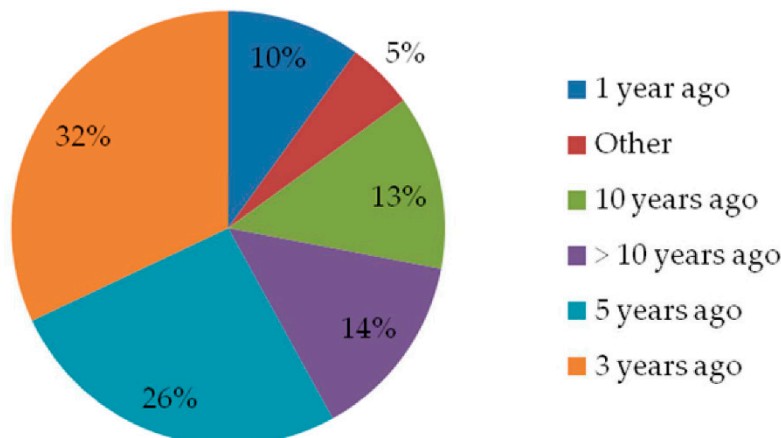

**Figure 5.** The years when households started using anthill soils in agriculture production.

For local maize varieties, the smallholder farmers reported that they applied anthill soil mostly to Gankanta (42%), Go by red (24%) and Mapongwe a Chitonga (23%). All these maize varieties are traditional landraces grown amongst the smallholder farmers of Southern Zambia and are a stop-gap measure to improved maize varieties if not accessed in time [37] and also perceived to be resilient to drought conditions [38], which ensures food security at household level. Smallholder farmers further explained that the use of anthill soil in agriculture production lies in the belief that this resource can increase yield (84%), improve soil fertility (63%) and contributed to enhancing household food security (49%). The anthill soil in the study districts is applied to legume crops as well.

Farmers also perceived that with use of anthill soils and planting legume crops such as cowpeas, beans and groundnuts, accrued benefits were more than when they used commercial fertilizer only. Amongst the respondents, 30% expected improved soil fertility, while 13% and 27% anticipated yield increase and improved household food security, respectively. In a similar study in Ghana, Ansong et al. [36] affirmed that the decision by farmers to use organic resources depended on the availability of the resource and general conditioning of the soil.

Generally, farmers indicated that with the use of anthill soil in agriculture production, benefits were more compared with conventional soil fertility management practices, with 87% and 68% respectively reporting high yields and cheap access to the resource as the major gains observed. A Chi-square test showed a significant association between smallholder farmers' start year of using anthill soil to improve soil fertility and access to credit schemes (Chi-Square ($\chi^2$) =12.616, DF = 6, *p*-value = 0.05) while there was no significant association on the use of commercially available fertilizer(Chi-Square ($\chi^2$) = 4.514, DF = 6, *p*-value = 0.0607).There was also significant association (Chi-Square ($\chi^2$) = 56.959, DF = 36, *p*-value = 0.015) between the start year of utilizing the anthill soil (from three years and beyond) and benefits accrued.

However, despite these reported benefits of anthill soils in agriculture production in the study districts, some farmers talked about the challenges faced, which included requirement of more water

by the resource (81%), inadequate labour (56%), handling and transportation (47%) and determining the required anthill soil quantities by the crop (17%)—with a paltry of respondents indicating that the resource requires less water standing at 6% and other reasons (2%) (Figure 6). Similar to farmers' observations, [39] indicated that the anthill soil requires more water because of its high suction characteristics compared with surrounding soil.

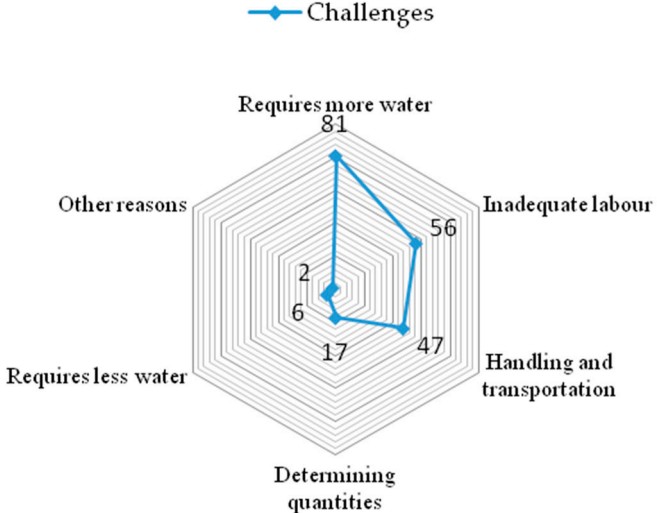

**Figure 6.** Radar chart showing challenges in anthill soil utilization.

In our study, we also found that smallholder farmers used vegetation type, i.e., species composition around anthill (85%), soil type (54%), other gestures (4%), size and shape of anthill as indicators for identifying appropriateness of anthill for soil improvement in agriculture production. We further established that the anthill identification is traditionally done by men (59%) while a combination of both men and women was represented by 36%. Men at a village level are normally considered key decision makers on agriculture activities and hence have a big share in the anthill identification aspect.

*3.9. Anthill Soil Fertility Management Calendar*

Smallholder farmers across the study districts reported that identification of anthill soil for use in crop production mostly commences in June according to 66% of the respondents, while 25% and 11% of the sampled households mentioned the months of July and August, respectively. Furthermore, farmers suggested that the underlying reason for starting anthill spotting during the stated month of June, is attributed to the condition of the anthill—which is slightly soft at the time. With regard to this aspect, however, 49% of the sampled households revealed that they start anthill digging in June, while 39% and 21% begin the process in July and August, respectively.

In terms of transporting the resource to the field, we found in the current study that the practice is mostly done in August (47%), followed by July (31%), September (25%), June (10%), October (8%), and November (3%), with December having a paltry representation of 1%. As for the participation in anthill soil application in the field, 81% reported that both men and women were involved while men alone contributed (13%), women alone (2%) and hired labour (10%). Given this scenario, we note in this study that women play a crucial role in anthill soil application. The application methods vary as well depending on the tillage system used by the farmers.

Farmer's Choice of Using Anthill Soils in Agriculture Production

An estimation of the multinomial logistic model for the determinants of smallholder farmer's choice of anthill soil utilization in their agriculture crop production systems showed that labour and limited capital/finance were the main drivers in the choice of this soil fertility management strategy. As for the biophysical factors (soil factors, weeds, pests and diseases), there is no influence on the

decision by smallholders to use anthill soil since no significant differences between the final and null model are observed. It is, however, expected that a female farmer is 19.91 and 20.29 times more likely to consider labour and limited capital/finance than the male counterpart in reference to other social economic factors before embarking on using anthill soil in agriculture production.

Of the two influencing variables (labour and limited capital/finance), only labour was found to be statistically significant at a 5% level. The −2Log likelihood ratio has an estimated figure of 19.80. This helped in obtaining the Chi-Square value, which normally informs on the goodness of fit for the model in comparison to the null model. The model containing the three variables (limited capital/finance, labour and others) was found to be significant at a 5% confidence interval, while the Cox and Snell R Square and Nagelkerke R Square is 0.022 and 0.044, respectively. Overall, we note that 90.8% of the respondents were predicted correctly (Table 4).

### 3.10. Farmer Issues to Agriculture Production

Key Benefits, Constraints and Coping Strategies to the Application of Anthill Soil Practices in Cropping Systems

Most farmers explained that anthill soil utilization has been beneficial to their agriculture production. Benefits disclosed lay in being cheap to access, improved crop yield, soil fertility improvement and food security while constraints reported focused on wilting of crops, germination challenges due to the hardness of the soil resulting in poor yield and labour intensiveness during collection of the resource and termite attacks. The discussants explained that these constraints itemized, were more pronounced during the drought periods and were in no way experienced in times of normal rain seasons.

The other major challenge mentioned as well is poor soil condition and limited access to extension services. As a result of this, most smallholder farmers talked about how they tended to use the anthill soil as an alternative to boost the fertility of the poor sandy soils, prevalent in the study areas. In support of the use of anthill soils in crop production, Haitao et al. [40] asserted that these soils are fertile due to the bioturbation activities which influence the chemical and physical characteristics of the soil. This is made possible through the ant building activities, where small residues are transported from a deeper layer on to the surface and organic matter is interred under, thereby changing the bulk density, particle constituents including the water holding capacity. However, to cope with the constraints that are more prominent during the drought conditions, farmers made it clear that they engage in ploughing, ripping/basin making, application of cattle manure, replanting, harrowing, irrigating and switch to alternative coping mechanisms which included trading and sale of livestock.

### 3.11. Farmer Access to Anthill Soil Technology Information

A significant part of the respondents (87%) indicated that their most important source of information on general agriculture practices was government extension workers, followed by neighbours (7%), radio/television (7%), own experience (6%), farmers organization (3%) with school/non-governmental organizations getting a paltry (2%). However, the major information gaps reported by farmers during focus group discussions related to anthill soil utilization lay in the aspects of limited extension or research services and lack of presence by non-governmental organizations, which play a key role in ensuring information that is disseminated to the rural communities on various agricultural practices. In this regard, the farmers called for the development of simple tools that could be used for anthill soil collection. Setting up of demonstrations on application methods in basins or ripped lines was reported to be key in disseminating the information on anthill soil technology utilization with emphasis on mechanization.

**Table 4.** Logistic Model Parameter Estimates: social economic factors for anthill soil utilization.

| Variable | | B | Standard Error | WaldTest | df | Sig. | Exp (β) | 95% Confidence Interval for Exp (β) | |
|---|---|---|---|---|---|---|---|---|---|
| | | | | | | | | **Lower Bound** | **Upper Bound** |
| Labour | Intercept | −1.216 | 0.403 | 9.131 | 1 | 0.003 | | | |
| | [Sex = female] | 19.905 | 1.071 | 345.565 | 1 | 0.000 | 441,088,960 | 54,087,124 | 3,597,149,460 |
| | [Sex = male] | 0 | | | 0 | | | | |
| Limited_capital | Intercept | 2.405 | 0.201 | 143.188 | 1 | 0.000 | | | |
| | [Sex = female] | 20.291 | 0.000 | | 1 | | 649,094,122 | 649,094,122 | 649,094,122 |
| | [Sex = male] | 0 | | | 0 | | | | |

**Model Summary**

| | |
|---|---|
| Model Chi-square | 8.844 |
| Model Sig. | 0.012 |
| −2 Log likelihood | 19.793 |
| Cox and Snell R Square | 0.022 |
| Nagelkerke R Square | 0.044 |
| % correct predictions | 90.8 |

Tadesse and Kwok [41] asserted that experience across many countries indicated that the adoption and spread of any technology call for adjustments in commitment and behavior of all key stakeholders—without which adoption of the promoted technologies becomes a challenge. This was also affirmed by [42] who observed that exposure of farmers to training increased their ability to internalize and use the information relevant to the agriculture technology at a farm level, which essentially leads to immense use and ensures sustainability of the technologies. Matata et al. [43] came to agreement with this point, stressing that extension contact is a very important aspect of changing the attitude amongst farmers towards adopting any technology being promoted. Cafer and Rikoon [44] stressed that the frequency of contact is considered a very important aspect of the smallholder farmer's ability to have access to any agriculture innovation earmarked for diffusion to them.

## 4. Conclusions and Recommendations

This study evaluated the traditional soil fertility management practice with the use of anthill soil in maize cropping systems with farmer experiences in Choma and Pemba districts of Southern Zambia. Our work has demonstrated that most smallholder farmers started using the anthill soils a minimum of three years ago. The main reason that forced the farmers to become captivated in the practice is largely in part due to social economic and biophysical factors—which amongst them include limited finance to procure mineral fertilizer and poor soil condition. Overall, we observe in this study that most of the smallholder farmers reported having key benefits accrued with the use of soils as fertilizer from specific anthills, and these lay in the belief that this resource can increase yield, improve soil fertility and contribute to enhanced household food security.

The smallholder farmers perceived that more benefits were expected with the use of anthill soils and planting legume crops such as cowpeas, beans and groundnuts. Additionally, the smallholder farmers indicated that with the use of anthill soil in agriculture production, benefits were more comparable with conventional soil fertility management practices, as the resource was easy to access. However, despite these reported benefits of anthill soils, some smallholder farmers explained that they faced challenges which included a requirement of more water by the resource, inadequate labour at the household level, handling and transportation and determining the required anthill soil quantities per unit of area such as a hectare. We noted that there was serious knowledge and research gaps, as most of the farmers relied on their fellow farmers to obtain information on efficient utilization of the anthill soil and general agriculture practices.

In light of the prevailing situation, we thus recommend the following action to be taken at different scales to tackle some of the constraints confronted by smallholder farmers in anthill soil utilization technology:

(i)   Institutional Level: in order to scale out the anthill soil technology, there is a need for strong linkages between the researchers and extension agents (private and government actors) for efficient application of anthill soil to be disseminated to farmers. In this study we found that most smallholders had limited knowledge on the best efficient methods of anthill soil technology practice and therefore conducting awareness messages and demonstrations would prove instrumental in this regard. This would most likely be more impactful in farmer field schools.

(ii)  Policy Level: policy makers should be made aware on how the anthill soil technology could be sustainably integrated in conservation agriculture production. Policy should stress the use of improved equipment for collection of the anthill soil, unlike the current trend where rudimental tools that are tenaciously strenuous and less efficient are still being employed by smallholder farmers.

By engaging in such actions, we aspire to address the challenges encountered in the use of anthill soil by smallholder farmers, which will contribute to enhanced soil fertility and eventually improved crop productivity and food security in general at community level.

**Author Contributions:** Conceptualization, K.C. and P.A.N.; methodology, K.C.; software, K.C.; validation, P.A.N. and E.M.; formal analysis, K.C.; investigation, K.C.; resources, P.A.N.; data curation, K.C.; writing—original draft preparation, K.C.; writing—review and editing, P.A.N.; visualization, K.C.; supervision, P.A.N. and E.M.; project administration, P.A.N.; funding acquisition, P.A.N., E.M. and K.C.

**Funding:** This study was funded by Centre for Research, Agricultural Advancement, Teaching Excellence and Sustainability in Food and Nutritional Security (CREATES-FNS) of The Nelson Mandela African Institution of Science and Technology (NM-AIST), Arusha, Tanzania (Credit No. 5799-TZ). The APC was funded by "The Pilot Programme for Climate Resilience (PPCR)", Ministry of National Planning and Development Lusaka, Zambia.

**Acknowledgments:** The authors would like to thank the Ministry of Agriculture, extension wing and the department of Zambia Agriculture Research Institute, Mochipapa Research Station, Choma for their technical support during data collection.

**Conflicts of Interest:** The authors declare no conflict of interest.

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
