# Peer review of "Socio-Economic Factors for Anthill Soil Utilization by Smallholder Farmers in Zambia"

_sustainability, doi:10.3390/su11184849_

Round 1

Reviewer 1 Report

Anthill Soil Utilization in Maize Cropping Systems as an Endogenous Soil Fertility Management Practice by Smallholder Farmers in Zambia

 (review)

 The article concerns a very important problem of improving soil fertility in Zambia. Farmers reported besides poor rainfall patterns (95%) also decreasing soil fertility (70%).

The study presents a very rich literature review - 42 bibliographic items. MATERIALS AND METHODS chapter is described comprehensively. The research process was planned and carried out correctly. The survey questionnaire was organized to cover questions on household demographic characteristics,  social  assets,  physical  assets,  human  assets,  natural  assets,  income  sources, site      characteristics, integrated soil fertility management practices and anthill soil utilization. Quantitative and qualitative analysis has been carried out. The results are presented in the form of clear figures and tables.

The literature and research carried out show that the described method can be useful in small farms in many countries of Africa and Asia, where there is a deficit of manure.

Remarks

Proposal: please change the title, e.g. “Socio-Economic Factors for Anthill Soil Utilization by Smallholder Farmers in Zambia”. Reason: the nature of the study was questionnaires. The Authors did not conduct chemical tests to assess soil fertility.

Please indicate the source for the Figure 1.

Chapter 2.1.1. Climate [line 119-130] is a compilation of an article from Wikipedia https://en.wikipedia.org/wiki/Choma,_Zambia

Please change it.

Reviewer 2 Report

Sustainability_555-1879

Comments

The paper reports the results of a survey regarding the use of anthill soil in small farming systems. The use of this soil as practice that increases the fertility of the surrounding soils is interesting and it is important for the region. My biggest concern relies on the length of the paper. Before resubmission, it should be shortened, so  I recommend major corrections.

In some parts of the Results and Discussion, the paper resembles more a technical report rather a research paper, but the paper is within the scope of the special issue.

Some specific comments are the following:

In MM there are some subsections that should be shortened. For example, 2.1.3 and 2.2.1 are too long.

Try to use another bibliographic source in figure 2.

Table 1 is not clear. It refers to percentage? What is HH size? Please check these values.

Table 2: USD. Provide the information in the table.

Line 353: the link of crop rotations and maize production (which is highlighted in the title) is not clear.
